# OpenReview forum: "Spurious Forgetting in Continual Learning of Language Models"
_ICLR.cc/2025/Conference — ICLR 2025 Poster_

### Official Review · Reviewer_rjuy · 2024-10-31

**Soundness:** 2
**Presentation:** 3
**Contribution:** 2
**Rating:** 5
**Confidence:** 4

**Summary:**

This paper studies the issue of forgetting during the training process of large language models (LLMs) and proposes a novel perspective on LLM forgetting. The authors categorize forgetting into two types: "decrease in task alignment capability" and "loss of knowledge." The decrease in task alignment capability is referred to as "pseudo-forgetting." When pseudo-forgetting occurs, the model’s capability can be restored with minimal data replay.
The authors also explore the pseudo-forgetting process at the model parameter level and find that, when aligning to different tasks, the orthogonal gradient directions of different tasks result in a loss of alignment with the original task when aligning with a new one. Furthermore, the authors study parameter changes across different model layers and find that the lower layers (embedding and first layer) play an essential role in task alignment. Freezing these layers can effectively alleviate the issue of pseudo-forgetting.

**Strengths:**

Proposes a novel perspective on catastrophic forgetting in LLM training, suggesting that LLMs do not lose knowledge but rather lose the ability to align with tasks.
Provides some theoretical support for the parameter-freezing training method through experiments.
Conducts thorough theoretical analysis. Experimental results across different methods are adequately explained, and a formalized definition of "pseudo-forgetting" is given.
The proposed method of freezing lower-layer parameters shows good performance.

**Weaknesses:**

1 Lack of Originality and Insufficient Literature Review:
The use of parameter freezing to mitigate forgetting in models is not a novel approach, and similar techniques have been explored previously. However, the authors do not provide an adequate discussion of prior research related to parameter freezing. This omission is significant, as a comparison with existing methods or a clear differentiation of this approach from prior studies would enhance the work's originality and position it within the context of established research.
2 Limited Evidence for Generalizability:
The experiments in the paper focus primarily on a synthetic biographical dataset, which raises questions about the broader applicability of the proposed parameter-freezing method. To demonstrate its effectiveness across diverse settings, it would be beneficial for the authors to test the method on additional, more varied datasets, including real-world datasets, and on a broader range of model architectures. Without this evidence, the method's generalizability remains uncertain.
3 Unclear Representation and Interpretation in Figures:
The figures, particularly Figure 3, are challenging to interpret, making it difficult to follow the intended findings. For instance, the paper later claims that data replay aligns the model with task 1 initially and then realigns with task 0, but this interpretation is not evident in Figure 3. The gradient descent path seems to diverge further from task 0, which contradicts the paper’s claims about realignment. Clearer visualizations or explanations of how the figures illustrate these alignment shifts are necessary for readers to understand the proposed method’s impact fully.

**Questions:**

1 Distinguishing Knowledge and Alignment in LLMs:
In the Introduction, the authors propose that an LLM’s task performance consists of both underlying knowledge and alignment capability. However, it’s unclear how these two aspects can be empirically distinguished. While task performance can be observed directly, what methods can effectively gauge the model’s internal knowledge versus its alignment with specific tasks? In Chapter 3, the authors describe two parameter update directions as reflecting "task 1 knowledge" and "task 0 alignment." Could the authors provide more clarity or empirical evidence to substantiate this differentiation? For instance, could ablation studies or other targeted analyses help isolate the effects of knowledge retention versus alignment?
2 Practical Value of "Pseudo-Forgetting" as a Concept:
The authors introduce "pseudo-forgetting" to describe cases where the model’s performance declines not due to actual knowledge loss, but rather due to a misalignment with the task at hand. Given that continual learning models must retain task alignment to be effective, what is the practical utility of distinguishing between true forgetting and pseudo-forgetting? Does this concept offer actionable insights for the design of more resilient continual learning strategies? If so, could the authors expand on the implications of pseudo-forgetting and how it might guide future methods for reducing task misalignment?
3 Initial Training Phase and Task Alignment Loss:
In Chapter 3, the authors extensively discuss how LLMs lose alignment with the original task in the first 150 steps of training on a new task, then progressively align with the new one. Could the authors clarify whether a warm-up phase was applied during this fine-tuning process? Alternatively, might the observed alignment shift at this stage be influenced by initial optimizer state instability, leading to greater parameter fluctuations rather than true alignment loss? Additionally, given that 150 steps might be insufficient to fully assess alignment shifts, could the authors investigate the effects of a longer initial phase or analyze optimizer state changes to clarify the nature of this alignment shift?

---

> ### Author Response · Authors · 2024-11-23
> **Response to Reviewer rjuy (Part 1/8)**
>
> > W1: Lack of Originality and Insufficient Literature Review: The use of parameter freezing to mitigate forgetting in models is not a novel approach, and similar techniques have been explored previously. However, the authors do not provide an adequate discussion of prior research related to parameter freezing. This omission is significant, as a comparison with existing methods or a clear differentiation of this approach from prior studies would enhance the work's originality and position it within the context of established research.
>
> Thank you for the insightful comment! While freezing layers is indeed a well-known concept, our work focuses on the identification and mitigation of *spurious forgetting* in continual learning of language models. The proposed *Freeze* strategy is intended as a simple yet effective baseline for addressing spurious forgetting. Importantly, our main contributions extend far beyond the introduction of the *Freeze* strategy. Here are the clarifications:
>
> 1. **About the Originality of the *Freeze* Strategy**:
>    - Our largest contribution is that we are the first to **identify and analyze the phenomenon of spurious forgetting in continual learning of language models** (`lines 078–079`).
>    - We discovered that spurious forgetting arises due to the loss of task alignment rather than the degradation of underlying knowledge (`lines 079–080`).
>    - A theoretical analysis of the causes of spurious forgetting was presented (`lines 080–081`).
>    - The *Freeze* strategy was proposed as a practical and effective method to mitigate this issue (`lines 081–082`).
>
>    **We want to emphasize that we did not invent freezing layers method; rather, we utilized this simple idea to effectively address the spurious forgetting problem we identified.** Our experiments showed that *Freeze* outperformed four well-known methods: EWC (a regularization-based method), LAMOL (a generative replay-based method), Task Vector (a model-merging-based method), and Gradient Projection (a gradient-based method). These results highlight the novelty and practical value of *Freeze* in continual learning for language models. **Therefore, we believe that our greatest contribution is identifying a new problem (spurious forgetting) and addressing it with a simple and effective approach.**
>
> 2. **Literature Review**:
>    In the latest manuscript, we expanded our discussion of parameter freezing in the related work section `Appendix A.3` to clarify its relation to continual learning. The new added part is shown below:
>      - Parameter freezing is a straightforward strategy for mitigating catastrophic forgetting. Architecture-based methods [R1-R4] can be considered a form of parameter freezing, as they typically train only a small proportion of parameters, such as LoRA [R1], prompts [R2-R4], or adapters [R5]. However, these methods generally capture less knowledge compared to full finetuning [R6]. Additionally, [R7-R8] propose freezing the backbone of LLMs and training only classifiers during continual learning, but their experiments are limited to classification tasks.
>
>      - Model expansion techniques [R9] also effectively prevent forgetting by freezing old layers and adding new layers for subsequent tasks. However, this approach is impractical for real-world applications due to the resource overhead of expanding the model for each new task.
>
>      - Unlike the parameter-freezing strategies discussed above, the proposed *Freeze* method can be applied to full finetuning of LLMs in real-world continual learning scenarios, such as alignment and continual instruction tuning. This distinguishes *Freeze* as a versatile and practical solution for addressing catastrophic forgetting in diverse settings.
>
> [R1] LoRAMoE: Alleviating World Knowledge Forgetting in Large Language Models via MoE-Style Plugin (ACL 2024)
>
> [R2] Progressive Prompts: Continual Learning for Language Models (ICLR 2023)
>
> [R3] Learning to Prompt for Continual Learning (CVPR 2022)
>
> [R4] CODA-Prompt: COntinual Decomposed Attention-based Prompting for Rehearsal-Free Continual Learning (CVPR 2023)
>
> [R5] Continual Sequence Generation with Adaptive Compositional Modules (ACL 2022)
>
> [R6] Lora learns less and forgets less (TMLR 2024)
>
> [R7] Learn or Recall? Revisiting Incremental Learning with Pre-trained Language Models (ACL 2024)
>
> [R8] Teamwork Is Not Always Good: An Empirical Study of Classifier Drift in Class-incremental Information
> Extraction (ACL 2023 Findings)
>
> [R9] LLaMA Pro: Progressive LLaMA with Block Expansion (ACL 2024)

---

> > ### Comment · Reviewer_rjuy · 2024-12-02
> > **"spurious forgetting"**
> >
> > I appreciate the clarification on the novelty of the "spurious forgetting" concept, I still feel that the response could benefit from a more detailed comparison with existing work, especially regarding the use of freezing layers. The authors reference several relevant studies, but I think the distinction between these existing methods and their Freeze strategy could be more explicitly drawn. For example, while the Freeze method is positioned as a versatile solution, it would help to show how it performs in comparison to other contemporary methods across a broader set of continual learning scenarios.

---

> ### Author Response · Authors · 2024-11-23
> **Response to Reviewer rjuy (Part 2/8)**
>
> > W2: Limited Evidence for Generalizability: The experiments in the paper focus primarily on a synthetic biographical dataset, which raises questions about the broader applicability of the proposed parameter-freezing method. To demonstrate its effectiveness across diverse settings, it would be beneficial for the authors to test the method on additional, more varied datasets, including real-world datasets, and on a broader range of model architectures. Without this evidence, the method's generalizability remains uncertain.
>
> **Response to W2**:
>
> We thank the reviewer for raising concerns about the generalizability of *Freeze*. In response, we conducted additional experiments involving finetuning on real-world datasets, including improving math ability with *MetaMathQA* and code ability with *Magicoder-Evol-Instruct-110K*. For evaluation, we used *GSM8K* and *Math* as benchmarks for math ability and *HumanEval* for code ability. To measure the forgetting of general ability, we considered a comprehensive set of benchmarks, including:
> - **Popular Aggregated Benchmarks**: *MMLU*, *BBH*
> - **World Knowledge**: *TriviaQA*
> - **Reading Comprehension**: *BoolQ*
> - **Commonsense Reasoning**: *HellaSwag*
>
> We recommend referring to the revised paper in `Appendix I.5.1` for detailed experimental settings. Below, we summarize the results from finetuning on *MetaMathQA* and *Magicoder-Evol-Instruct-110K*:
>
> **1. Math Ability**: Finetuning on *MetaMathQA* for one epoch (`Table 35` in `Appendix I.5`)
>
> |                                                    |              |        |                 |        |          |        |           |         |
> |----------------------------------------------------|--------------|--------|-----------------|--------|----------|--------|-----------|---------|
> |                                                    | Math Ability |        | General Ability |        |          |        |           |         |
> |                                                    | Math         | GSM8K  | MMLU            | BBH    | TriviaQA | BoolQ  | HellaSwag | Average |
> | LLaMa-3-8B-Instruct                                | 27.48        | 79.00  | 68.28           | 52.87  | 64.78    | 84.34  | 74.19     | 68.89   |
> | LLaMa-3-8B-Instruct+SFT(math,lr=2e-5)              | 27.30        | 73.92  | 54.28           | 9.05   | 28.94    | 8.81   | 12.20     | 22.66   |
> | LLaMa-3-8B-Instruct+SFT(math,lr=5e-6)              | 31.26        | 80.29  | 65.67           | 42.85  | 61.77    | 78.65  | 71.83     | 64.15   |
> | LLaMa-3-8B-Instruct+SFT(math,lr=5e-6,Freeze)       | 31.43        | 80.17  | 66.84           | 47.53  | 62.56    | 81.35  | 72.27     | 66.11   |
> | Qwen2.5-7B-Instruct                                | 51.54        | 80.52  | 74.25           | 65.95  | 55.92    | 83.46  | 81.18     | 72.15   |
> | Qwen2.5-7B-Instruct+SFT(math,lr=5e-6)              | 53.30        | 79.27  | 73.73           | 68.99  | 56.77    | 80.55  | 82.71     | 72.55   |
> | Qwen2.5-7B-Instruct+SFT(math,lr=5e-6,Freeze)       | 53.79        | 80.12  | 74.11           | 69.35  | 58.14    | 81.67  | 82.65     | 73.18   |
> | Mistral-8B-Instruct-2410                           | 36.52        | 80.38  | 65.86           | 64.88  | 61.06    | 84.65  | 83.41     | 71.97   |
> | Mistral-8B-Instruct-2410+SFT(math,lr=5e-6)         | 39.73        | 81.50  | 59.15           | 54.02  | 45.91    | 60.59  | 67.41     | 57.42   |
> | Mistral-8B-Instruct-2410+SFT(math,lr=5e-6,Freeze)  | 39.82        | 81.77  | 62.18           | 58.39  | 53.84    | 71.57  | 73.81     | 63.96   |

---

> ### Author Response · Authors · 2024-11-23
> **Response to Reviewer rjuy (Part 3/8)**
>
> **2. Code Ability**: Finetuning on *Magicoder-Evol-Instruct-110K* for one epoch (`Table 36` in `Appendix I.5`)
>
> |                                                    |              |                 |        |          |        |           |         |
> |----------------------------------------------------|--------------|-----------------|--------|----------|--------|-----------|---------|
> |                                                    | Code Ability | General Ability |        |          |        |           |         |
> |                                                    | HumanEval    | MMLU            | BBH    | TriviaQA | BoolQ  | HellaSwag | Average |
> | LLaMa-3-8B-Instruct                                | 59.15        | 68.28           | 52.87  | 64.78    | 84.34  | 74.19     | 68.89   |
> | LLaMa-3-8B-Instruct+SFT(code,lr=2e-5)              | 60.37        | 65.34           | 58.64  | 63.42    | 85.47  | 70.56     | 68.69   |
> | LLaMa-3-8B-Instruct+SFT(code,lr=5e-6)              | 62.20        | 68.08           | 66.50  | 64.91    | 85.08  | 72.80     | 71.47   |
> | LLaMa-3-8B-Instruct+SFT(code,lr=5e-6,Freeze)       | 62.33        | 68.14           | 67.61  | 64.77    | 85.15  | 73.24     | 71.78   |
> | Qwen2.5-7B-Instruct                                | 82.32        | 74.25           | 65.95  | 55.92    | 83.46  | 81.18     | 72.15   |
> | Qwen2.5-7B-Instruct+SFT(code,lr=5e-6)              | 83.10        | 73.99           | 70.40  | 58.01    | 86.21  | 82.31     | 74.18   |
> | Qwen2.5-7B-Instruct+SFT(code,lr=5e-6,Freeze)       | 83.22        | 74.58           | 71.21  | 57.38    | 86.52  | 82.84     | 74.51   |
> | Mistral-8B-Instruct-2410                           | 72.56        | 65.86           | 64.88  | 61.06    | 84.65  | 83.41     | 73.75   |
> | Mistral-8B-Instruct-2410+SFT(code,lr=5e-6)         | 75.81        | 64.15           | 68.83  | 60.21    | 86.27  | 83.72     | 72.64   |
> | Mistral-8B-Instruct-2410+SFT(code,lr=5e-6,Freeze)  | 76.34        | 64.64           | 67.38  | 61.54    | 86.53  | 84.60     | 72.94   |
> |                                                    |              |                 |        |          |        |           |         |
>
>
> The additional experiments provide robust evidence for the generalizability of *Freeze* across multiple real-world datasets, diverse domains (math and code), and varied LLM architectures. For a detailed analysis of the experimental results, please refer to `Appendix I.5.2` of the revised manuscript. We hope these new results address your concerns!

---

> ### Author Response · Authors · 2024-11-23
> **Response to Reviewer rjuy (Part 4/8)**
>
> > W3: Unclear Representation and Interpretation in Figures: The figures, particularly Figure 3, are challenging to interpret, making it difficult to follow the intended findings. For instance, the paper later claims that data replay aligns the model with task 1 initially and then realigns with task 0, but this interpretation is not evident in Figure 3. The gradient descent path seems to diverge further from task 0, which contradicts the paper’s claims about realignment. Clearer visualizations or explanations of how the figures illustrate these alignment shifts are necessary for readers to understand the proposed method’s impact fully.
>
> **Response to W3**:
>
> Thank you for your valuable comment! We believe there may be some misunderstanding regarding `Figure 3`. Below, we provide clarifications about the axes and trajectory in the figure to address your concerns:
>
> ### **Clarification of Axes in Figure 3**
> As stated in `lines 203–205`, the landscape in `Figure 3` represents a two-dimensional space defined by weight update directions during different training stages:
> - **Y-axis**: The weight update direction during the **initial 150 steps**, which corresponds to the process of undoing *Task 0 alignment*.
> - **X-axis**: The weight update direction during the **subsequent steps**, which corresponds to the process of learning *Task 1 knowledge*.
>
> This decomposition highlights the distinct learning processes in these two stages. For further clarification, please refer to our detailed explanation in `Response to Q1-2`.
>
> ### **Interpretation of the Trajectory in the Data Replay Setting**
> The trajectory in the data replay setting (`Figure 3(b)`) can be interpreted as follows:
> 1. **Beginning of the Trajectory**: Initially, the trajectory moves in the opposite direction of *Task 0 alignment* (Y-axis) during the first 150 steps. This phase corresponds to the *undoing of Task 0 alignment*.
> 2. **Middle of the Trajectory**: The trajectory starts to shift along the direction of *Task 1 knowledge* (X-axis) while still moving in the opposite direction of *Task 0 alignment*. This is because data replay progressively encourages *Task 0 re-alignment* throughout the learning process of Task 1, as the model searches for a balance between *Task 1 alignment* and *Task 0 alignment* (illustrated in `Figure 6`).
> 3. **Final Phase**: The trajectory continues to move in the direction of *Task 1 knowledge* (X-axis) while gradually finding a common direction for *Task 0 alignment* and *Task 1 alignment*. This final phase reflects the model's attempt to reconcile alignment for both tasks.
>
>
> ### **Revised Explanation in the Manuscript**
> To address potential misunderstandings, we have added additional explanations about the axes and trajectory in `Figure 3` in the revised manuscript. Additonally, we add interpretation of the trajectory in the data replay setting in `Appendix G.2`. These updates clarify how the trajectory reflects the dynamic interplay between *Task 0 re-alignment* and *Task 1 knowledge* during data replay.
>
> We believe these changes make the interpretation more accessible and consistent with the proposed method’s impact. We hope this resolves your concerns and improves the clarity of `Figure 3`.

---

> ### Author Response · Authors · 2024-11-23
> **Response to Reviewer rjuy (Part 5/8)**
>
> > Q1-1: Distinguishing Knowledge and Alignment in LLMs: In the Introduction, the authors propose that an LLM’s task performance consists of both underlying knowledge and alignment capability. However, it’s unclear how these two aspects can be empirically distinguished. While task performance can be observed directly, what methods can effectively gauge the model’s internal knowledge versus its alignment with specific tasks?
>
> **Response to Q1-1**:
>
> Thank you for your insightful comment! Recent studies have explored the relationship between underlying knowledge and task alignment in LLMs. Below, we summarize key findings that address your concern:
>
> 1. **Recent Studies about "Underlying Knowledge vs. Task Performance"**:
>    Recent works [R10-13] demonstrate that LLMs may fail to exhibit strong task performance despite possessing the necessary underlying knowledge. These studies provide methods to probe internal knowledge by intervening in hidden states. For example:
>    - **Safety Alignment and Knowledge Retention**:
>      - [R10] shows that finetuning safety-aligned LLMs (e.g., GPT-3.5-Turbo and LLaMa-2-7B-Chat) on benign instruction-following datasets significantly increases the harmfulness rate. This suggests that safety alignment suppresses harmful behavior without removing the underlying harmful knowledge, which can be elicited by further finetuning.
>    - **Mechanistic Understanding of Alignment**:
>      - [R11] investigates alignment algorithms like DPO [R12] and finds that they introduce offsets in the residual stream to avoid toxic outputs. By reversing these offsets, the model reverts to pre-aligned behavior, revealing retained knowledge.
>    - **Truthfulness of LLMs**:
>      - [R13] finds that LLMs often produce falsehoods even when they internally know the correct answer. Intervening during inference can elicit truthful responses.
>
> 2. **Gap Between Knowledge and Alignment in Real-World Scenarios**:
>    Studies also highlight scenarios where task performance improves by eliciting latent knowledge:
>    - For example, adding the prompt *Let's think step by step* significantly enhances performance on math problems [R14], implying that chain-of-thought techniques help LLMs better utilize their underlying knowledge to solve math problems.
>    - Similarly, GPT-4 may yield incorrect answers in one context while providing correct answers in another, underscoring the role of task alignment in utilizing latent knowledge.
>
> These findings suggest that underlying knowledge is not equivalent to task performance in LLMs and that alignment plays a critical role in eliciting knowledge effectively.
>
> [R10] Fine-tuning Aligned Language Models Compromises Safety, Even When Users Do Not Intend To! (ICLR 2024)
>
> [R11] A Mechanistic Understanding of Alignment Algorithms: A Case Study on DPO and Toxicity (ICML 2024)
>
> [R12] Direct preference optimization: Your language model is secretly a reward model (NIPS 2023)
>
> [R13] Inference-Time Intervention: Eliciting Truthful Answers from a Language Model (NIPS 2023)
>
> [R14] Large language models are zero-shot reasoners (NeurIPS 2022)

---

> ### Author Response · Authors · 2024-11-23
> **Response to Reviewer rjuy (Part 6/8)**
>
> > Q1-2: In Chapter 3, the authors describe two parameter update directions as reflecting "task 1 knowledge" and "task 0 alignment." Could the authors provide more clarity or empirical evidence to substantiate this differentiation? For instance, could ablation studies or other targeted analyses help isolate the effects of knowledge retention versus alignment?
>
>
> **Response to Q1-2**:
>
> Thank you for your insightful comment! We believe there may be some misunderstanding regarding the experimental setup in `Section 3.1`. **Our carefully designed experiments already isolate the effects of knowledge retention and alignment.** Below is a detailed explanation:
>
> 1. **Experimental Decomposition**:
>    - As illustrated in `Figure 2(b)`, when LLMs are finetuned on *Task 0*, they primarily learn *Task 0 alignment* because the underlying knowledge for Task 0 is acquired during pretraining.
>    - The performance gain for Task 0 (solid green line in `Figure 2(a)`) reflects the outcome of learning *Task 0 alignment*.
>    - After 150 steps of finetuning on *Task 1*, Task 0 performance drops sharply from nearly 100% to 10%. However, the *recovered Task 0 accuracy* remains close to 100% after 150 steps, indicating that the model loses *Task 0 alignment* but retains the underlying Task 0 knowledge.
>    - As Task 1 accuracy improves from 0% to nearly 100% during subsequent finetuning steps, the model learns *Task 1 knowledge*.
>
> This separation of learning processes is explicitly designed in our experiments and shown in `Figure 2(a)`.
>
> 2. **Supporting Evidence**:
>    - `Figure 3` and `Figure 4` further validate the decomposition of *Task 1 knowledge* and *Task 0 alignment*. For example, the orthogonal updates in weight space (shown in `Figure 4`) highlight the separation between these two learning processes.
>
> In summary, the experimental design in `Section 3.1` effectively decomposes the learning of knowledge and alignment. Performance changes in `Figure 2(a)` and additional evidence in `Figures 3` and `4` clearly substantiate our claims. We hope this explanation clarifies your concerns.
>
> ---
>
> > Q2-1: Practical Value of "Pseudo-Forgetting" as a Concept: The authors introduce "pseudo-forgetting" to describe cases where the model’s performance declines not due to actual knowledge loss, but rather due to a misalignment with the task at hand. Given that continual learning models must retain task alignment to be effective, what is the practical utility of distinguishing between true forgetting and pseudo-forgetting?
>
>
> **Response to Q2-1**:
>
> Thank you for your insightful comment!
>
> We believe that distinguishing between actual knowledge loss and the loss of task alignment is crucial, especially in the era of LLMs. Here are the key reasons:
>
> 1. **Real-World Importance**:
>    - In safety finetuning, it is essential to identify whether harmful knowledge within an LLM has been erased or merely suppressed. Misinterpreting suppressed knowledge as erased could lead to unsafe deployments.
>    - Similarly, if task alignment is lost while underlying knowledge is retained, performance on old tasks can often be recovered with simple techniques, as demonstrated in the *recovery experiments* in our paper (`Section 2`).
>
> 2. **Theoretical and Empirical Significance**:
>    - Recent works (discussed in `Response to Q1-1`) share a similar decomposition view between underlying knowledge and task alignment. We are the first to highlight this distinction in the context of continual learning and demonstrate that it leads to spurious forgetting in language models.
>    - Misinterpreting spurious forgetting as catastrophic forgetting of knowledge could result in the development of ineffective algorithms that aim to recover forgotten knowledge unnecessarily, rather than addressing alignment issues directly.
>
> 3. **Practical Utility Demonstrated**:
>    - To further show the practical value of identifying spurious forgetting, we have added new experiments on real-world datasets. Details are provided in `Response to W2`.

---

> ### Author Response · Authors · 2024-11-23
> **Response to Reviewer rjuy (Part 7/8)**
>
> > Q2-2: Does this concept offer actionable insights for the design of more resilient continual learning strategies? If so, could the authors expand on the implications of pseudo-forgetting and how it might guide future methods for reducing task misalignment?
>
> **Response to Q2-2**:
>
> Thank you for your valuable comment! Below, we summarize actionable insights provided in our paper (`Section 5`) and new experiments on real-world scenarios (`Appendix I.5`) that underscore the practical utility of the pseudo-forgetting concept.
>
> ### **1. Actionable Insights from `Section 5: Solution to Spurious Forgetting`**
> Our findings provide actionable insights for designing more resilient continual learning strategies:
> - **Ineffectiveness of Existing Strategies**:
>   Popular continual learning methods such as regularization, generative replay, model merging, and gradient projection are ineffective for addressing spurious forgetting (`Section 5.1`).
>
> - **Effectiveness of Data Replay**:
>   Data replay mitigates spurious forgetting by encouraging the re-alignment of Task 0 (`Section 5.2`). This suggests that avoiding the undoing of alignment can help retain old task performance (as described in `lines 451–453` and `lines 67–69`).
>
> - **Freezing Bottom Layers**:
>   Freezing the bottom layers of the model is a highly effective strategy for alleviating spurious forgetting. This approach works well on both synthetic datasets (`Section 5.2`) and real-world datasets (`Section 5.3`).
>
> ### **2. Additional Experiments on Real-World Scenarios in `Appendix I.5`**
> To emphasize the value of identifying spurious forgetting, we conducted additional experiments on real-world datasets to showcase the utility of the proposed freezing strategy. Details of these experiments are provided in `Response to W2`.
>
> We hope these insights and experiments demonstrate how distinguishing pseudo-forgetting can guide future methods for reducing task misalignment and improving the resilience of continual learning models.
>
>
> ---
>
>
> > Q3-1: Initial Training Phase and Task Alignment Loss: In Chapter 3, the authors extensively discuss how LLMs lose alignment with the original task in the first 150 steps of training on a new task, then progressively align with the new one. Could the authors clarify whether a warm-up phase was applied during this fine-tuning process?
>
> **Response to Q3-1**:
>
> Thank you for your valuable comment! We did not use a warm-up strategy during fine-tuning. As described in `Figure 2(b)` and `lines 1187–1189`, the initial learning rate was set to $5 \times 10^{-6}$, and a cosine learning rate decay was applied (from $5 \times 10^{-6}$ to $4.5 \times 10^{-6}$). We believe that whether or not using warm-up strategy does not affect the spurious forgetting because $5 \times 10^{-6}$ is a small learning rate. We provide more evidence in `Response to Q3-2` to support our claim.

---

> ### Author Response · Authors · 2024-11-23
> **Response to Reviewer rjuy (Part 8/8)**
>
> > Q3-2: Alternatively, might the observed alignment shift at this stage be influenced by initial optimizer state instability, leading to greater parameter fluctuations rather than true alignment loss? Additionally, given that 150 steps might be insufficient to fully assess alignment shifts, could the authors investigate the effects of a longer initial phase or analyze optimizer state changes to clarify the nature of this alignment shift?
>
> **Response to Q3-2**:
>
> Thank you for your thoughtful comments! We understand the concern that the rapid performance decline in the first 150 steps might be due to instability at the beginning of training. Below, we address this concern from three aspects:
> 1. **Existing evidence from the loss landscape**
> 2. **Existing evidence under various continual learning settings**
> 3. **New experiments on different combinations of learning rates and optimizers (detailed in `Appendix G.1.4`)**
>
> ### **1. Evidence from the Loss Landscape**
> As shown in the loss landscape in `Figure 3(a)`, during the first 150 steps:
> - The trajectory moves directly in the direction that decreases the *Task 1 loss* most rapidly. This indicates that the optimizer functions effectively at the beginning of fine-tuning, reducing the *Task 1 loss* from approximately 4.0 to 2.0.
> - In contrast, the *Task 0 loss* increases rapidly, explaining the observed performance decline for Task 0 during the initial phase of fine-tuning.
>
> This demonstrates that the optimizer’s state is stable and effectively finds the gradient direction for the current task (Task 1).
>
> ### **2. Evidence Across Continual Learning Settings**
> The observed performance drop for the old task in the initial training steps occurs across various continual learning scenarios, as detailed in `Appendix G`. For example:
> - **Different Task Type and Number of Tasks**:
>   In `Figure 8`, performance on the old task drops within the first 40 steps when fine-tuning subsequent tasks. Similarly, in `Figure 11`, a performance decline is observed within the first 200 steps.
> - **Different Training Samples**:
>   `Figures 9` and `10` show that spurious forgetting occurs regardless of training samples, except when the number of examples in the new task is very small (e.g., 20 or 200 instances).
>
> These findings confirm that spurious forgetting consistently occurs at the beginning of fine-tuning across diverse experimental conditions.
>
> ### **3. New Experiments on Learning Rates and Optimizers (`Appendix G.1.4`)**
> To further investigate whether spurious forgetting is caused by instability at the start of fine-tuning, we conducted experiments using different optimizers (AdamW and SGD) and initial learning rates ranging from $1 \times 10^{-4}$ to $1 \times 10^{-7}$. Key findings include:
> - **SGD**:
>   - Converges much slower than AdamW. For example, with a learning rate of $1 \times 10^{-5}$ or smaller, Task 0 training loss does not decrease even after 10,000 steps.
> - **AdamW**:
>   - Effectively reduces training loss with learning rates as low as $5 \times 10^{-7}$.
>
> The results (detailed in `Figure 12`, `Appendix G.1.4`) indicate that spurious forgetting occurs at different steps depending on the optimizer and learning rate:
> - Using SGD, spurious forgetting appears after approximately 3,000 steps ($1 \times 10^{-4}$) and 6,000 steps ($5 \times 10^{-5}$).
> - With AdamW, spurious forgetting occurs after 200, 600, and 1,000 steps for learning rates of $5 \times 10^{-6}$, $1 \times 10^{-6}$, and $5 \times 10^{-7}$, respectively.
>
> These results confirm that spurious forgetting is **not** due to initial instability during fine-tuning but rather reflects inherent dynamics of the learning process.
>
> We hope this response clarifies the nature of the alignment shift during the initial phase of training. The additional evidence provided here and in `Appendix G.1.4` demonstrates that spurious forgetting is a consistent phenomenon unrelated to optimizer instability.

---

> > ### Comment · Reviewer_rjuy · 2024-12-02
> >
> > In the case of very small sample sizes (e.g., 20 or 200 examples), the authors observed reduced spurious forgetting. Could further analysis be provided on how task difficulty and sample size interact to influence the extent of spurious forgetting, particularly for tasks with more complex or hierarchical structures?

---

> > > ### Author Response · Authors · 2024-12-02
> > > **Response to Reviewer rjuy**
> > >
> > > > Comment 2: In the case of very small sample sizes (e.g., 20 or 200 examples), the authors observed reduced spurious forgetting. Could further analysis be provided on how task difficulty and sample size interact to influence the extent of spurious forgetting, particularly for tasks with more complex or hierarchical structures?
> > >
> > > **Response to Comment 2:**
> > >
> > > Thank you for this insightful comment! We observed no spurious forgetting when sample sizes are extremely small (e.g., 20 or 200 individuals in our experimental setting), as demonstrated in `Figure 9`. The underlying reasons are analyzed below:
> > >
> > > 1. **Task Size and Model Capability:**
> > >    When the new tasks are significantly smaller or simpler compared to the model's capacity (e.g., pretraining on 100K individuals), the model can easily adapt to the new tasks by transferring existing knowledge without requiring large updates. For example, as illustrated in the trajectory of Task 1's loss in `Figure 3(a)`, the model requires large updates to adapt to Task 1.
> > >
> > > 2. **Task Alignment Perspective:**
> > >    From the perspective of task alignment, a common direction between the task alignments of new and old tasks can be easily identified when the sample size is small or the task is simple. This will correspond to a larger intersection area in `Figure 6`.
> > >
> > > 3. **When Spurious Forgetting Occurs?**
> > >    Spurious forgetting is more likely to arise when new tasks are sufficiently challenging or have an adequate number of training samples relative to the model's existing capacity. In such scenarios, learning a common direction for task alignment across tasks becomes less trivial, potentially leading to task misalignment and forgetting.
> > >
> > > ### **Planned Manuscript Update:**
> > > We will include the above analysis in `Appendix G.1.2` in the camera-ready version if accepted, as we are currently unable to update the manuscript. This addition will provide further clarity on how task difficulty and sample size influence spurious forgetting in models with varying capabilities.

---

> > ### Comment · Reviewer_rjuy · 2024-12-02
> >
> > In the current experiments, spurious forgetting is observed early in the training process. Would the authors consider extending the training phase further to assess whether the forgetting stabilizes or worsens with continued training, particularly in scenarios where the model is exposed to a large number of tasks?

---

> > > ### Author Response · Authors · 2024-12-02
> > > **Response to Reviewer rjuy**
> > >
> > > > Comment 3: In the current experiments, spurious forgetting is observed early in the training process. Would the authors consider extending the training phase further to assess whether the forgetting stabilizes or worsens with continued training, particularly in scenarios where the model is exposed to a large number of tasks?
> > >
> > > **Response to Comment 3:**
> > >
> > > ### **1. Training with More Steps**
> > >
> > > We have conducted additional experiments to assess the behavior of spurious forgetting over an extended training phase. Our findings show that **spurious forgetting stabilizes early in training and remains consistent thereafter.** Specifically:
> > >
> > > - **Stability of Forgetting:**
> > >   The **gap between Task 0 ACC** and **Recovered Task 0 ACC** remains consistent from step 150 through to the final step (62,500), as shown in our experiments. Furthermore, in additional experiments not included in the manuscript, we extended training to over **300,000 steps** and observed that spurious forgetting remained stable. The key difference was that the decline in **Recovered Task 0 ACC** was more pronounced with the extended training.
> > >
> > > - **Sufficient Training Steps:**
> > >   The training duration (62,500 steps) in our experiments is adequate. As evidenced by the **loss landscape** in `Figure 3(a)`, the **Task 1 loss stabilizes** in a minimum basin, which suggests that further extending the training phase would not significantly impact the observed dynamics of spurious forgetting.
> > >
> > > Thus, based on our extended experiments, we conclude that the observed spurious forgetting is **stable over time**.
> > >
> > >
> > > ### **2. Training with a Larger Number of Tasks**
> > >
> > > We have evaluated the model’s behavior in **scenarios involving continual learning over five tasks**, with the results summarized in `Figure 8`. Key findings include:
> > >
> > > - **Consistent Spurious Forgetting:**
> > >   Spurious forgetting remains **consistent across all five tasks**.
> > >
> > > - **Evolution into True Forgetting:**
> > >   As new tasks are introduced, the **recovered performance on earlier tasks deteriorates more significantly** after each new task is learned. This indicates that spurious forgetting can evolve into **true forgetting** of knowledge from prior tasks.
> > >
> > > While our current experiments focus on five tasks, the observed trends in spurious forgetting and recovered performance suggest that these dynamics are **likely to hold in larger-scale continual learning scenarios** for LLMs.
> > >
> > > We recognize that experiments involving a much larger number of tasks (e.g., 100 tasks) would provide further insights into the dynamics of spurious forgetting. However, due to the **high computational cost** of training LLMs on such a large scale, we have opted to focus on a **practical and realistic scenario** involving five tasks. We believe this still provides valuable insights into the challenges of spurious forgetting in LLMs and continual learning.
> > >
> > > We hope this clarifies your question! Considering the approaching deadline, please let us know if you have any follow-up concerns. We sincerely hope you can consider our reply in your assessment and that we can further address any unclear explanations and remaining concerns, if any.

---

> > > ### Author Response · Authors · 2024-12-02
> > > **Response to Reviewer rjuy**
> > >
> > > Dear Reviewer rjuy,
> > >
> > > We hope this message finds you well!
> > >
> > > As today is the final day for reviewers to submit feedback, we kindly ask if you have any further questions or concerns. We sincerely hope that our responses have addressed your points, and we would greatly appreciate it if you could consider them in your assessment. If any explanations remain unclear or if there are still issues that need further attention, we would be more than happy to provide additional clarifications.
> > >
> > > Thank you once again for your time and valuable feedback.
> > >
> > > Best regards,
> > > Authors

---

> ### Author Response · Authors · 2024-12-02
> **Response to Reviewer rjuy**
>
> > Comment 1: I appreciate the clarification on the novelty of the "spurious forgetting" concept, I still feel that the response could benefit from a more detailed comparison with existing work, especially regarding the use of freezing layers. The authors reference several relevant studies, but I think the distinction between these existing methods and their Freeze strategy could be more explicitly drawn. For example, while the Freeze method is positioned as a versatile solution, it would help to show how it performs in comparison to other contemporary methods across a broader set of continual learning scenarios.
>
> **Response to Comment 1**:
>
> Thank you for your thoughtful comment and for highlighting the importance of drawing clearer distinctions between our Freeze strategy and prior work. Below, we provide further clarification:
>
> ## **1. Unique Contribution of Freezing Bottom Layers:**
> We highlight that **no prior work proposes freezing bottom layers as a standalone solution** for continual learning in LLMs. This novelty forms the core of our Freeze strategy and distinguishes it from existing approaches.
>
> ## **2. Comparisons with Relevant Works:**
> **While freezing-based strategies have been explored in prior works ([R7-R9]), the scope, methodology, and applicability differ significantly. Therefore, these works [R7-R9] are not comparable to the Freeze strategy in our continual learning settings**:
>
> - **[R7-R8] Freezing the Entire Backbone LLM:**
>   - **Approach:**
>     These studies propose freezing the entire backbone LLM while **training only classifiers** for continual tasks, such as **continual text classification.**
>   - **Distinction:**
>     - **Our Focus:** Our work addresses a broader and more practical range of continual learning scenarios for LLMs, including:
>       - **Safety alignment**
>       - **Continual instruction tuning**
>       - **Continual knowledge editing**
>       - **Instance incremental learning**
>     - In contrast, [R7-R8] focus narrowly on classification tasks, which are **not applicable to the generative tasks** shown in our paper.
>
> - **[R9] Model Expansion (LLaMA Pro):**
>   - **Approach:**
>     [R9] explores model expansion, adding new Transformer layers to adapt LLMs for specific domains (e.g., code and math tasks).
>   - **Distinction:**
>     - Unlike Freeze, this approach modifies the model architecture rather than optimizing learning dynamics.
>     - Model expansion strategies are **less relevant to continual learning**, as they focus on accommodating new tasks by increasing model size rather than addressing forgetting or ensuring scalability.

---

### Official Review · Reviewer_zFfn · 2024-11-02

**Soundness:** 3
**Presentation:** 4
**Contribution:** 3
**Rating:** 8
**Confidence:** 3

**Summary:**

The paper analyzes the phenomenon of catastrophic forgetting in the continual learning of language models. The authors argue that this forgetting stems from a decline in task alignment rather than a loss of knowledge. They support their claim through: (1) constructing a biography dataset to demonstrate their findings from the perspectives of performance, loss landscape, model weights, and features; (2) conducting a theoretical analysis of orthogonal updates in model weights. Then they revisit existing methods and propose a straightforward and effective freezing strategy to mitigate forgetting, which outperforms other methods.

**Strengths:**

1. The paper presents a novel and compelling claim regarding the phenomenon of forgetting in continual learning.
2. It adopts multiple perspectives to effectively substantiate this claim.
3. The paper provides a clear theoretical analysis based on the observation of orthogonal updates in model weights.
4. The proposed method, grounded in the theoretical analysis and observation, outperforms existing baseline methods.

**Weaknesses:**

The support for the assumption that the weight perturbation matrix lies in the left null space of the weight matrix in Assumption 4.4 appears to be simply based on the observations from Figure 4. However, in Figure 4, orthogonal updates occur only after the first 150 steps in the model's bottom layers. Based on performance observations, the accuracy of Task 0 has already significantly declined during these initial 150 steps. This does not support the conclusion that ``"observed performance declines are largely a result of orthogonal updates in model weights."``

I would appreciate it if the authors could address the concern or clarify if I have misunderstood any part of the paper.

**Questions:**

See above.

---

> ### Author Response · Authors · 2024-11-23
> **Response to Reviewer zFfn (Part 1/1)**
>
> > **W1: The support for the assumption that the weight perturbation matrix lies in the left null space of the weight matrix in Assumption 4.4 appears to be simply based on the observations from Figure 4. However, in Figure 4, orthogonal updates occur only after the first 150 steps in the model's bottom layers. Based on performance observations, the accuracy of Task 0 has already significantly declined during these initial 150 steps. This does not support the conclusion that "observed performance declines are largely a result of orthogonal updates in model weights."**
>
> **Response to W1**:
> Thank you for your thoughtful comment! We believe there may be some misunderstanding regarding the concept of *orthogonal updates* as presented in our paper. Below, we clarify the points and address your concerns:
>
> ### **1. Clarification: What are orthogonal updates?**
> The *orthogonal updates* shown in `Figure 4` do **not** refer to a specific training stage, such as *after the first 150 steps*. As described in `lines 260–261` and shown in the legend of `Figure 4(b)`, the term *orthogonal updates* (orange color in `Figure 4(b)`) refers to the near-orthogonality between **two learning stages**:
> 1. **The first 150 steps** (primarily associated with undoing Task 0 alignment).
> 2. **Subsequent steps** (from step 150 to the end, primarily associated with learning Task 1 knowledge and alignment).
>
> ### **2. The role of orthogonal updates in spurious forgetting**
> - While *orthogonal updates* are a key cause of spurious forgetting, they do **not** directly lead to Task 0 performance declines. Instead:
>   - Because the direction of *undoing Task 0 alignment* opposes that of *learning Task 0 alignment*, and the model learns *Task 1 knowledge* and *Task 1 alignment* simultaneously from step 150 to the end, ***orthogonal updates* lead to *orthogonality in alignment* between Task 0 and Task 1**. This phenomenon is described in `lines 258–265` and illustrated in `Figure 6`.
>   - This *orthogonality in alignment* prevents the model from maintaining performance on Task 0, or from re-aligning to Task 0, as would occur with methods like data replay (`lines 451–453`). This misalignment is what we term as *spurious forgetting*.
> - The observed Task 0 performance declines during the first 150 steps are directly caused by the process of **undoing Task 0 alignment**. This is evidenced by the blue color in `Figure 4(b)`, representing the loss of alignment during this stage.
>
> To clarify, the statement in `line 318`—"observed performance declines are largely a result of orthogonal updates in model weights"—may have led to confusion. In the revised manuscript, we have updated it to:
> *"Observed spurious forgetting is largely a result of orthogonal updates in model weights."*
>
> ### **3. Additional clarification about Assumption 4.4**
> **Assumption 4.4 is not solely based on observations from `Figure 4`**. The orthogonality of model updates between the two learning stages (first 150 steps and subsequent steps) is further supported by the loss landscape in `Figure 3(a)`:
> - In `Figure 3(a)`, the trajectory of updates during *subsequent steps* (from step 150 to the end) is nearly orthogonal to the direction of Task 0 alignment.
> - This trajectory includes both *Task 1 knowledge* and *Task 1 alignment*. **The fact that the updates for Task 1 alignment do not overlap with Task 0 alignment implies *orthogonality in alignment* between Task 0 and Task 1.**
>
> We hope these clarifications resolve any misunderstandings.

---

> > ### Comment · Reviewer_zFfn · 2024-11-24
> >
> > Thank you for your detailed clarification. Based on Proposition 4.9, would freezing the first layer of the model also be effective in mitigating cumulative drift, since it similarly reduces the number of layers?

---

> > > ### Author Response · Authors · 2024-11-25
> > > **Response to Reviewer zFfn**
> > >
> > > We deeply appreciate your thoughtful and engaging feedback! Below, we address your question from both **theoretical** and **empirical** perspectives:
> > >
> > > #### 1. **Theoretical Perspective**:
> > >    - As stated in `Assumption 4.4` (*Perturbation on Weight Matrices*), if **all layers** update in the left null-space, freezing the first (topmost) layer can indeed have a similar effect as freezing the bottom (lowest) layer, as both actions reduce the number of layers involved in updates, according to `Proposition 4.9`.
> > >    - However, as demonstrated in `Figures 4(b) and 14(c)`, orthogonality is most prominent *only in the bottom layers* (e.g., the bottom 6 layers). This means that in real-world scenarios, where only the *bottom layers* satisfy the assumptions of `Assumption 4.4`, it would be more effective to freeze the *bottom layers* to mitigate cumulative drift.
> > >    - We have added a discussion about *freezing the topmost layers* in `Remark F.4` to better demonstrate the connection between `Proposition 4.9` and the proposed strategy of freezing the bottom layers.
> > >
> > > #### 2. **Empirical Perspective**:
> > >    - To validate this claim, we conducted an additional experiment in which we freeze the topmost layers, using the same experimental setup as in the Biography dataset experiment shown in `Table 1`. The results are summarized below:
> > >
> > > |                       | Task 0 Acc | Task 1 Acc |
> > > |-----------------------|------------|------------|
> > > | SEQ                   | 11.18      | 99.91      |
> > > | Freeze (Top 1 Layers) | 11.68      | 99.93      |
> > > | Freeze (Top 2 Layers) | 11.39      | 99.93      |
> > > | Freeze (Top 3 Layers) | 10.78      | 99.92      |
> > >
> > >    - The results show that **freezing the topmost layers does not result in significant improvement** for SEQ, which is consistent with the theoretical discussion. This further supports the idea that freezing the **bottom layers** is more effective in mitigating cumulative drift, as hypothesized.
> > >
> > > We hope this clarifies your question! Considering the approaching deadline, please let us know if you have any follow-up concerns. We sincerely hope you can consider our reply in your assessment and that we can further address any unclear explanations and remaining concerns, if any.

---

> > > > ### Comment · Reviewer_zFfn · 2024-11-25
> > > >
> > > > Thank you for your response. I believe the explanation that "orthogonality is most prominent only in the bottom layers" makes sense. However, I suggest that the authors clarify this point more explicitly in Remark 4.10 rather than relegating it to the appendix. The reasoning provided in Remark 4.10 —"since it reduces the number of layers L"—is insufficient.

---

> ### Author Response · Authors · 2024-11-25
> **Response to Reviewer zFfn**
>
> Thank you for your valuable suggestion! We have revised **Remark 4.10** in the latest version of the manuscript to clarify the points raised. The modified part is highlighted in **bold** below:
>
> - **Remark 4.10**:
>   `Proposition 4.9` shows that the bound of the final shift is proportional to $L (1 + \delta)^{L-1}$, indicating that the output is particularly sensitive to the number of layers $L$. This is reasonable because the shift accumulates from the bottom to the top layers. **Additionally, orthogonality is most prominent in the bottom layers (see `Figure 4b`), meaning that in real-world scenarios, only the bottom layers are likely to satisfy `Assumption 4.4`. This suggests that freezing the bottom layers may help mitigate the accumulated shift by reducing the number of layers that contribute to the shift in the output. Further discussion on this is provided in `Remark F.4`.**
>
> We hope this revision addresses your concern! Given the approaching deadline, please let us know if you have any further questions or follow-up concerns. We would be happy to clarify any remaining points and hope our revisions meet your expectations.

---

> ### Comment · Reviewer_zFfn · 2024-11-25
>
> I still suggest that the authors should further refine and strengthen the rigor of their theoretical proofs in subsequent revisions (e.g., addressing points raised by other reviewers, such as the observation that "nearly orthogonal updates primarily occur in the lower layers and have relatively small angles"). The authors should be more transparent in acknowledging the gap between their theoretical assumptions and actual experimental observations, e.g., discuss this in the limitations section. In my opinion, this gap is acceptable.
>
> Overall, the authors' response has addressed my concerns regarding the theoretical aspects, and I have raised my score accordingly.

---

> ### Author Response · Authors · 2024-11-26
> **Response to Reviewer zFfn**
>
> Thank you for your thoughtful comments and for raising the score! We truly appreciate your time and effort in reviewing our paper and engaging in the discussion.
>
> ### **1. Addressing the Theoretical Rigor and Experimental Observations**
>
> In response to your suggestion for further refining and strengthening the rigor of our theoretical proofs, we have made several updates to the manuscript. Specifically, we have added `Corollary F.5` in the Appendix, which provides a more rigorous theoretical result that addresses the observation about nearly orthogonal updates occurring **primarily in the lower layers**.
>
> - `Corollary F.5` is based on `Assumption F.4`, which assumes that orthogonal updates happen only in the bottom $L_{\text{bottom}}$ layers of the network.
> - `Corollary F.5` demonstrates that freezing the bottom $L_{\text{freeze}}$ layers (where $L_{\text{freeze}} \leq L_{\text{bottom}}$) effectively *reduces the bound on the shift in the final output*.
>
> We believe that these additions provide a clearer theoretical foundation that directly connects the orthogonal updates in the lower layers to the final output shift, as observed experimentally.
>
> ### **2. Addressing the Gap Between Assumptions and Empirical Observations**
>
> - To bridge the gap between our theoretical assumptions and actual experimental observations, we have revised `Remark 10` and `Remark F.6` to provide a more transparent discussion. These remarks now help readers better understand how the theoretical assumptions align with our empirical findings.
>
> We hope that these revisions help clarify the theoretical framework and its connection to our experimental results. We have carefully considered your feedback and believe these updates strengthen the rigor of our theoretical analysis. We encourage you to review the latest manuscript for the detailed content and the proof of `Corollary F.5`.
>
> Once again, thank you for your valuable feedback and support in improving the manuscript.

---

### Official Review · Reviewer_RhW8 · 2024-11-03

**Soundness:** 2
**Presentation:** 3
**Contribution:** 2
**Rating:** 6
**Confidence:** 3

**Summary:**

The paper explores the phenomenon of "spurious forgetting" in continual learning for language models. It challenges the assumption that performance degradation across sequential tasks implies actual knowledge loss, suggesting that many performance drops arise from suboptimal task alignment. The authors propose a new method, "Freeze," which involves freezing the lower layers of a model to reduce this issue.

**Strengths:**

1. The paper brings a novel perspective by distinguishing between knowledge forgetting and spurious forgetting, addressing a misconception in continual learning research. This distinction enhances our understanding of model behavior across sequential tasks, highlighting the significance of this work.

2. The work is generally well-articulated and of good quality, with well-structured arguments and detailed explanations, making the complex ideas accessible to a broad audience.

3. The "Freeze" approach is simple yet effective, offering a practical solution for reducing spurious forgetting. Its ease of implementation is particularly noteworthy.

**Weaknesses:**

1. Freezing layers for fine-tuning has long been a common practice and is not a novel approach. For instance, freezing lower layers was explored in the VGG paper [1].

2. The empirical results in this work do not provide strong evidence to support the assumptions related to orthogonality.

3. The proposed method is only compared to the SEQ method in real-world scenarios, which is a pretty weak baseline.

[1] Simonyan, K., & Zisserman, A. (2014). VERY DEEP CONVOLUTIONAL NETWORKS FOR LARGE-SCALE IMAGE RECOGNITION. arXiv preprint arXiv:1409.1556.

**Questions:**

1. Why are Task 1 knowledge and Task 0 alignment used as the bases for plotting the loss landscapes?

2. The theoretical framework relies primarily on a linear model. Could this assumption of linearity be too strong to realistically capture the complexities of the problem? What are the limitations of using a linear model?

3. Why isn’t exact match accuracy used as the evaluation metric for all tasks?

4. From a model weight perspective, why is the angle computed based on the column spaces rather than directly from the weight updates?

---

> ### Author Response · Authors · 2024-11-23
> **Response to Reviewer RhW8 (Part 1/8)**
>
> > W1: Freezing layers for fine-tuning has long been a common practice and is not a novel approach. For instance, freezing lower layers was explored in the VGG paper [1].
> >
> > [1] Simonyan, K., & Zisserman, A. (2014). VERY DEEP CONVOLUTIONAL NETWORKS FOR LARGE-SCALE IMAGE RECOGNITION. arXiv preprint arXiv:1409.1556.
>
>
> **Response to W1**:
> Thank you for the insightful comment! While freezing layers is indeed a well-known concept, such as in VGG [1], our work focuses on the identification and mitigation of *spurious forgetting* in continual learning of language models. The proposed *Freeze* strategy is intended as a simple yet effective baseline for addressing spurious forgetting. Importantly, our main contributions extend far beyond the introduction of the *Freeze* strategy. Here are the clarifications:
>
> 1. **About the Novelty of the *Freeze* Strategy**:
>    - Our largest contribution is that we are the first to **identify and analyze the phenomenon of spurious forgetting in continual learning of language models** (`lines 078–079`).
>    - We discovered that spurious forgetting arises due to the loss of task alignment rather than the degradation of underlying knowledge (`lines 079–080`).
>    - A theoretical analysis of the causes of spurious forgetting was presented (`lines 080–081`).
>    - The *Freeze* strategy was proposed as a practical and effective method to mitigate this issue (`lines 081–082`).
>
>    **We want to emphasize that we did not invent freezing layers method; rather, we utilized this simple idea to effectively address the spurious forgetting problem we identified.** Our experiments showed that *Freeze* outperformed four well-known methods: EWC (a regularization-based method), LAMOL (a generative replay-based method), Task Vector (a model-merging-based method), and Gradient Projection (a gradient-based method). These results highlight the novelty and practical value of *Freeze* in continual learning for language models. **Therefore, we believe that our greatest contribution is identifying a new problem (spurious forgetting) and addressing it with a simple and effective approach.**
>
> 2. **Connection to Related Work**:
>    In the latest manuscript, we expanded our discussion of parameter freezing in the related work section `Appendix A.3` to clarify its relation to continual learning. The new added part is shown below:
>      - Parameter freezing is a straightforward strategy for mitigating catastrophic forgetting. Architecture-based methods [R1-R4] can be considered a form of parameter freezing, as they typically train only a small proportion of parameters, such as LoRA [R1], prompts [R2-R4], or adapters [R5]. However, these methods generally capture less knowledge compared to full finetuning [R6]. Additionally, [R7-R8] propose freezing the backbone of LLMs and training only classifiers during continual learning, but their experiments are limited to classification tasks.
>
>      - Model expansion techniques [R9] also effectively prevent forgetting by freezing old layers and adding new layers for subsequent tasks. However, this approach is impractical for real-world applications due to the resource overhead of expanding the model for each new task.
>
>      - Unlike the parameter-freezing strategies discussed above, the proposed *Freeze* method can be applied to full finetuning of LLMs in real-world continual learning scenarios, such as alignment and continual instruction tuning. This distinguishes *Freeze* as a versatile and practical solution for addressing catastrophic forgetting in diverse settings.
>
> [R1] LoRAMoE: Alleviating World Knowledge Forgetting in Large Language Models via MoE-Style Plugin (ACL 2024)
>
> [R2] Progressive Prompts: Continual Learning for Language Models (ICLR 2023)
>
> [R3] Learning to Prompt for Continual Learning (CVPR 2022)
>
> [R4] CODA-Prompt: COntinual Decomposed Attention-based Prompting for Rehearsal-Free Continual Learning (CVPR 2023)
>
> [R5] Continual Sequence Generation with Adaptive Compositional Modules (ACL 2022)
>
> [R6] Lora learns less and forgets less (TMLR 2024)
>
> [R7] Learn or Recall? Revisiting Incremental Learning with Pre-trained Language Models (ACL 2024)
>
> [R8] Teamwork Is Not Always Good: An Empirical Study of Classifier Drift in Class-incremental Information
> Extraction (ACL 2023 Findings)
>
> [R9] LLaMA Pro: Progressive LLaMA with Block Expansion (ACL 2024)

---

> ### Author Response · Authors · 2024-11-23
> **Response to Reviewer RhW8 (Part 2/8)**
>
> > W2: The empirical results in this work do not provide strong evidence to support the assumptions related to orthogonality.
>
> **Response to W2**:
>
> Thanks for your valuable comment! We note that the orthogonality discussed in our work refers to the *alignment between tasks (Task 0 and Task 1)*, as illustrated in `Figure 6`. Here is a summary of the evidence supporting our claims:
>
> 1. **Evidence from Loss Landscapes (`Figures 3, 12`)**:
>    - In `Figure 3(a)`, the trajectory of updates after step 150 is nearly orthogonal to the direction of *Task 0 alignment*. Recall that trajectory of updates after step 150 includes the learning process of both *task 1 knowledge* and *task 1 alignment*. This demonstrates that the learning of *Task 1 alignment* does not overlap significantly with *Task 0 alignment*, resulting in orthogonal alignment between the tasks.
>
> 2. **Evidence from Weight Updates (`Figures 4, 13`)**:
>     - In `Figure 4(b)`, the updates in the two learning stages (first 150 steps vs. subsequent steps) show near-orthogonality. The orange color demonstrates this observation, consistent with the loss landscape results.
>     - Besides, `Figure 4(b)` shows the *the first 150 steps* causes the performance declines on Task 0 by undoing Task 0 Alignment (`lines 258-260`).
>     - In summary, the *orthogonal updates* in `Figure 4` lead to the *orthogonality in alignment* between Task 0 and Task 1 (illustruated in `Figure 6`). Due to the *orthogonality in alignment* between Task 0 and Task 1, the model fail to utilize the underlying knowledge of Task 0 (or fail to re-align to Task 0 like in data replay,`lines 451-453`) which is exactly the phenomona we termed *spurious forgetting*.

---

> > ### Comment · Reviewer_RhW8 · 2024-11-25
> > **Official Comment by Reviewer RhW8**
> >
> > Thank the authors for their clarification and additional results.
> >
> > As for W2, Figure 4(b) indicates that most angles concentrate below 45 degrees, which makes the conclusion of near-orthogonality unsolid. Hope the authors can provide some clarification on this.

---

> > > ### Author Response · Authors · 2024-11-25
> > > **Response to Reviewer RhW8**
> > >
> > > Thank you for raising this concern in W2! We appreciate the opportunity to clarify.
> > >
> > > ### **1. Clarification on Near-Orthogonality:**
> > >
> > > - As mentioned in `lines 263-265`, the near-orthogonal weight updates in the **bottom layers** contribute to the differences in alignment between Task 0 and Task 1, which in turn leads to the **spurious forgetting** observed in Task 0.
> > >
> > > - **We would like to emphasize that we did NOT claim near-orthogonal updates occur in ALL layers.** In fact, these near-orthogonal updates are specifically observed in the **bottom layers**. To avoid confusion, we also clarify that `Assumption 4.4`, which assumes orthogonal updates occur across all layers, is a simplified model used for theoretical analysis.
> > >
> > > - Our theoretical analysis, as presented in `Proposition 4.9` and `Remark 4.10`, shows that **the shift accumulates from the bottom to the top layers.** In other words, **while the near-orthogonal updates occur primarily in the bottom layers in real-world scenarios, the shift itself originates in the bottom layers and propagates upward** (as visualized in `Figure 5`).
> > >
> > > ### **2. Additional Clarification with Visualization:**
> > >
> > > - We also recommend referring to **Figure 14(c)**, where we visualize the angles of model weights in different components, including *dense_h_to_4h*, *attention.dense*, *attention.query*, *attention.key*, and *attention.value*. The results in **Figure 14(c)** demonstrate that the near-orthogonality is most prominent in the **bottom layers** of the model, further supporting our conclusions.
> > >
> > > We hope this additional clarification addresses your concern. Please let us know if you have any further questions or if there are any remaining points we can clarify.

---

> > > > ### Comment · Reviewer_RhW8 · 2024-11-27
> > > > **Official Comment by Reviewer RhW8**
> > > >
> > > > Thanks for the clarifications. I will adjust my score.

---

> > > > > ### Author Response · Authors · 2024-11-27
> > > > > **Response to Reviewer RhW8**
> > > > >
> > > > > Thank you for your feedback and for adjusting your score! We are delighted to hear that our clarifications have addressed your concerns. Your thoughtful review and support are greatly appreciated.

---

> ### Author Response · Authors · 2024-11-23
> **Response to Reviewer RhW8 (Part 3/8)**
>
> > **W3: The proposed method is only compared to the SEQ method in real-world scenarios, which is a pretty weak baseline.**
>
> **Response to W3**:
>
> Thank you for your valuable comment! To address this concern, we have added new experiments on real-world scenarios. However, we first want to clarify that Sequential Finetuning (SEQ) is not a weak baseline for continual learning with LLMs, especially on real-world datasets. In computer vision, SEQ is often considered the lower-bound performance for continual learning. In contrast, for LLMs, SEQ is a practical and widely used baseline for adapting models to new tasks when using small learning rates (e.g., 5e-6 in our paper). You can find the results of our new experiments on math and code datasets in `Appendix I.5` of the revised manuscript. Below are detailed points supporting SEQ as a practical baseline:
>
> ### **1. SEQ is NOT a weak baseline for continual learning with LLMs on real-world datasets:**
>
> - **Industry Practice**:
>     - Supervised finetuning (SFT) of pretrained LLMs on high-quality user-conversation data is a standard practice to improve instruction-following capabilities [R10-R12]. This process can be viewed as a two-task continual learning scenario where the first task is pretraining, and the second is finetuning on high-quality data.
>     - SFT, equivalent to SEQ, preserves most general ability obtained during pretraining when using small learning rates (e.g., 5e-6). For example, general ability may drop from 75% to 70% after finetuning. In contrast, in computer vision, SEQ typically results in significant performance degradation on old tasks (e.g., from 90% to 10%).
>
> - **Academic Advances**:
>     - Recent studies in continual learning with LLMs [R13-17] highlight the strong anti-forgetting ability of LLMs under SEQ. For example:
>         - [R13] shows that BERT experiences minimal forgetting when adapting to new tasks.
>         - [R14] extends these findings to broader scenarios, including continual text classification, intent classification, relation extraction, and named entity recognition. Their experiments cover various LLM architectures, sizes, and settings, such as task-incremental and class-incremental learning.
>         - [R15] investigates continual domain-adaptive training and reports a forgetting rate of just 1.05% for SEQ (Table 2).
>         - [R16] demonstrates that full finetuning outperforms LoRA in accuracy and sample efficiency for code and math. Specifically, finetuning LLaMa-2-7B on Magicoder-Evol-Instruct-110K and MetaMathQA results in a modest general ability drop from 67% to 60% after one epoch. For continual pretraining, forgetting on math tasks is negligible.
>         - [R17] explores continual instruction tuning, where LLMs are sequentially finetuned on eight diverse generation tasks. Their findings (Table 2) show low forgetting rates for SEQ: 2.58% for LLaMa-2-7B, 13.32% for LLaMa-2-13B, 2.89% for BaiChuan2-7B-Instruct, 2.77% for Vicuna-7B-V1.5, and 9.27% for Vicuna-13B-V1.5.
>
> These findings, along with the new results in our revised paper, reinforce that SEQ is a practical baseline for continual learning with LLMs, especially in real-world scenarios.
>
>
> [R10] GPT-4 Technical Report
>
> [R11] Stanford alpaca: An instruction-following llama model.
>
> [R12] Vicuna: An Open-Source Chatbot Impressing GPT-4 with 90%* ChatGPT Quality
>
> [R13] Can BERT Refrain from Forgetting on Sequential Tasks? A Probing Study (ICLR 2023)
>
> [R14] Learn or Recall? Revisiting Incremental Learning with Pre-trained Language Models (ACL 2024)
>
> [R15] Continual pre-training of language models (ICLR 2023)
>
> [R16] Lora learns less and forgets less (TMLR 2024)
>
> [R17] TRACE: A Comprehensive Benchmark for Continual Learning in Large Language Models

---

> ### Author Response · Authors · 2024-11-23
> **Response to Reviewer RhW8 (Part 4/8)**
>
> ### **2. New Experiments on Real-World Scenarios**
>
> We conducted additional experiments involving finetuning on real-world datasets, including improving math ability with *MetaMathQA* and code ability with *Magicoder-Evol-Instruct-110K*. For evaluation, we used *GSM8K* and *Math* as benchmarks for math ability and *HumanEval* for code ability. To measure the forgetting of general ability, we considered a comprehensive set of benchmarks, including:
> - **Popular Aggregated Benchmarks**: *MMLU*, *BBH*
> - **World Knowledge**: *TriviaQA*
> - **Reading Comprehension**: *BoolQ*
> - **Commonsense Reasoning**: *HellaSwag*
>
> We recommend referring to the revised paper in `Appendix I.5.1` for detailed experimental settings. Below, we summarize the results from finetuning on *MetaMathQA* and *Magicoder-Evol-Instruct-110K*:
>
> **(1) Math Ability**: Finetuning on *MetaMathQA* for one epoch (`Table 35` in `Appendix I.5`)
>
> |                                                    |              |        |                 |        |          |        |           |         |
> |----------------------------------------------------|--------------|--------|-----------------|--------|----------|--------|-----------|---------|
> |                                                    | Math Ability |        | General Ability |        |          |        |           |         |
> |                                                    | Math         | GSM8K  | MMLU            | BBH    | TriviaQA | BoolQ  | HellaSwag | Average |
> | LLaMa-3-8B-Instruct                                | 27.48        | 79.00  | 68.28           | 52.87  | 64.78    | 84.34  | 74.19     | 68.89   |
> | LLaMa-3-8B-Instruct+SFT(math,lr=2e-5)              | 27.30        | 73.92  | 54.28           | 9.05   | 28.94    | 8.81   | 12.20     | 22.66   |
> | LLaMa-3-8B-Instruct+SFT(math,lr=5e-6)              | 31.26        | 80.29  | 65.67           | 42.85  | 61.77    | 78.65  | 71.83     | 64.15   |
> | LLaMa-3-8B-Instruct+SFT(math,lr=5e-6,Freeze)       | 31.43        | 80.17  | 66.84           | 47.53  | 62.56    | 81.35  | 72.27     | 66.11   |
> | Qwen2.5-7B-Instruct                                | 51.54        | 80.52  | 74.25           | 65.95  | 55.92    | 83.46  | 81.18     | 72.15   |
> | Qwen2.5-7B-Instruct+SFT(math,lr=5e-6)              | 53.30        | 79.27  | 73.73           | 68.99  | 56.77    | 80.55  | 82.71     | 72.55   |
> | Qwen2.5-7B-Instruct+SFT(math,lr=5e-6,Freeze)       | 53.79        | 80.12  | 74.11           | 69.35  | 58.14    | 81.67  | 82.65     | 73.18   |
> | Mistral-8B-Instruct-2410                           | 36.52        | 80.38  | 65.86           | 64.88  | 61.06    | 84.65  | 83.41     | 71.97   |
> | Mistral-8B-Instruct-2410+SFT(math,lr=5e-6)         | 39.73        | 81.50  | 59.15           | 54.02  | 45.91    | 60.59  | 67.41     | 57.42   |
> | Mistral-8B-Instruct-2410+SFT(math,lr=5e-6,Freeze) | 39.82        | 81.77  | 62.18           | 58.39  | 53.84    | 71.57  | 73.81     | 63.96   |

---

> ### Author Response · Authors · 2024-11-23
> **Response to Reviewer RhW8 (Part 5/8)**
>
> **(2) Code Ability**: Finetuning on *Magicoder-Evol-Instruct-110K* for one epoch (`Table 36` in `Appendix I.5`)
>
> |                                                    |              |                 |        |          |        |           |         |
> |----------------------------------------------------|--------------|-----------------|--------|----------|--------|-----------|---------|
> |                                                    | Code Ability | General Ability |        |          |        |           |         |
> |                                                    | HumanEval    | MMLU            | BBH    | TriviaQA | BoolQ  | HellaSwag | Average |
> | LLaMa-3-8B-Instruct                                | 59.15        | 68.28           | 52.87  | 64.78    | 84.34  | 74.19     | 68.89   |
> | LLaMa-3-8B-Instruct+SFT(code,lr=2e-5)              | 60.37        | 65.34           | 58.64  | 63.42    | 85.47  | 70.56     | 68.69   |
> | LLaMa-3-8B-Instruct+SFT(code,lr=5e-6)              | 62.20        | 68.08           | 66.50  | 64.91    | 85.08  | 72.80     | 71.47   |
> | LLaMa-3-8B-Instruct+SFT(code,lr=5e-6,Freeze)       | 62.33        | 68.14           | 67.61  | 64.77    | 85.15  | 73.24     | 71.78   |
> | Qwen2.5-7B-Instruct                                | 82.32        | 74.25           | 65.95  | 55.92    | 83.46  | 81.18     | 72.15   |
> | Qwen2.5-7B-Instruct+SFT(code,lr=5e-6)              | 83.10        | 73.99           | 70.40  | 58.01    | 86.21  | 82.31     | 74.18   |
> | Qwen2.5-7B-Instruct+SFT(code,lr=5e-6,Freeze)       | 83.22        | 74.58           | 71.21  | 57.38    | 86.52  | 82.84     | 74.51   |
> | Mistral-8B-Instruct-2410                           | 72.56        | 65.86           | 64.88  | 61.06    | 84.65  | 83.41     | 73.75   |
> | Mistral-8B-Instruct-2410+SFT(code,lr=5e-6)         | 75.81        | 64.15           | 68.83  | 60.21    | 86.27  | 83.72     | 72.64   |
> | Mistral-8B-Instruct-2410+SFT(code,lr=5e-6,Freeze)  | 76.34        | 64.64           | 67.38  | 61.54    | 86.53  | 84.60     | 72.94   |
> |                                                    |              |                 |        |          |        |           |         |
>
>
> The additional experiments provide robust evidence for the generalizability of *Freeze* across multiple real-world datasets, diverse domains (math and code), and varied LLM architectures. For a detailed analysis of the experimental results, please refer to `Appendix I.5.2` of the revised manuscript. We hope these new results address your concerns!
>
>
> ### **3. New Experiments on Comparing with LAMOL and EWC**
>
> We conducted additional experiments to further compare LAMOL and EWC with the Freeze strategy. We note that Task Vector requires saving multiple checkpoints and search for best hyper-parameters using these checkpoints and thus not applicable in real-world setting. Besides, Gradient Projection shows every poor performance on Biography dataset and thus not used for further comparison. The result in `Table 10` are shown as below:
>
> |                            |                       |                         |                         |                         |
> |----------------------------|-----------------------|-------------------------|-------------------------|-------------------------|
> | Scenario                   | CKE                   |                         | IIL                     |                         |
> | Metric                     | Efficacy ($\uparrow$) | Paraphrase ($\uparrow$) | Mem. Acc.  ($\uparrow$) | Gen. Acc.  ($\uparrow$) |
> | SEQ                        | 62.47±0.49            | 58.24±0.53              | 35.98±0.17              | 12.61±0.14              |
> | EWC                        | 64.17±0.43            | 64.17±0.63              | 37.19±0.56              | 11.58±0.30              |
> | LAMOL                      | 66.84±0.35            | 61.17±0.32              | 38.84±0.16              | 12.66±0.28              |
> | Freeze (1 layers, 1 task)  | 70.88±0.69            | 64.19±0.96              | 37.00±0.23              | 13.06±0.10              |
> | Freeze (2 layers, 1 task)  | 70.65±0.45            | 68.60±0.35              | 42.18±0.05              | 14.19±0.21              |
> |                            |                       |                         |                         |                         |
>
>
> The result shows that Freeze outperforms LAMOL and EWC by a large margin, consistent to the results on Biography dataset.
>
> The additional experiments provide additional evidence that Freeze outperforms representative continual learning methods in real-world datasets. We hope these new results address your concerns!

---

> ### Author Response · Authors · 2024-11-23
> **Response to Reviewer RhW8 (Part 6/8)**
>
> > **Q1: Why are Task 1 knowledge and Task 0 alignment used as the bases for plotting the loss landscapes?**
>
> **Response to Q1**:
>
> Thank you for your thoughtful comment! It seems there may be some misunderstanding about the loss landscapes in `Figure 3`. To clarify, we will (1) explain what the axes represent in `Figure 3` and (2) provide the rationale for choosing these axes.
>
> ### **1. What are the bases (axes)?**
> As described in `lines 203-205` of the manuscript, the landscape in `Figure 3` represents a two-dimensional space defined by weight update directions from different stages of the training process:
> - The **y-axis** corresponds to the weight update direction during the initial 150 steps of finetuning.
> - The **x-axis** corresponds to the weight update direction during subsequent finetuning steps.
>
> These axes are directly tied to the decomposition of the learning process into two components: **undoing Task 0 alignment** and **learning Task 1 knowledge**. Specifically:
> - The **y-axis** represents the direction associated with changes in *Task 0 alignment*.
> - The **x-axis** represents the direction associated with the acquisition of *Task 1 knowledge*.
>
> ### **2. Why were these bases (axes) chosen?**
> The axes were selected because the carefully designed experimental setup (`lines 139-140`) allows us to decompose the learning process in the Sequential Finetuning Setting (SEQ) into two distinct stages, as shown in `Figure 2(a)`:
>   - (1) **First 150 Steps**: During these steps, the model primarily focuses on **undoing Task 0 alignment**.
>       - As illustrated in `Figure 2(b)`, the LLM only learns *Task 0 alignment* when finetuned on Task 0 since all task knowledge was already obtained during pretraining.
>       - The solid green line in `Figure 2(a)` shows performance gains in Task 0, which result solely from learning Task 0 alignment.
>       - After 150 steps of finetuning on Task 1, Task 0 performance drops sharply from nearly 100% to 10%. However, the recovered Task 0 accuracy remains near 100%, indicating that the LLM only loses *Task 0 alignment* during these steps.
>
>   - (2) **Subsequent Steps**: During this phase, the model focuses on learning **Task 1 knowledge**.
>      - As seen in `Figure 2(a)`, Task 1 accuracy rises from 0% to nearly 100% during these steps, demonstrating the LLM's acquisition of Task 1 knowledge.
>
> This decomposition into two stages provides a clear framework for analyzing the loss landscape, making these axes a natural choice for understanding the interplay between Task 0 alignment and Task 1 knowledge during sequential finetuning.
>
>
> To address potential misunderstandings, we have added further explanations of `Figure 3` in the revised manuscript. We hope these updates, along with the above clarification, address your concerns fully.

---

> ### Author Response · Authors · 2024-11-23
> **Response to Reviewer RhW8 (Part 7/8)**
>
> > **Q2-1: The theoretical framework relies primarily on a linear model. Could this assumption of linearity be too strong to realistically capture the complexities of the problem?**
>
> **Response to Q2-1**:
>
> Thank you for your insightful comment. We acknowledge the concern and believe that our assumption of stacked linear models with residual connections effectively captures the crucial learning dynamics of the spurious forgetting problem. Here are the key reasons:
>
> ### **1. Simplified Model Choice**
> We chose a simplified model (stacked linear model with residual connections) because:
> - **Orthogonal updates are more closely tied to the number of layers rather than specific components like self-attention and MLPs.** As shown in `Figure 13`, the angles between weight updates are largely independent of specific components but strongly correlated with the number of layers.
> - To avoid unnecessary complexity in analyzing specific model designs (e.g., self-attention), we opted for this simplification. This approach aligns with recent studies on Transformers, such as using single-layer models [R18] or diagonal weight matrices [R19]. Hence, employing a simplified model is reasonable if it effectively reflects the dynamics of the targeted problem.
>
> ### **2. Empirical Consistency with Theoretical Results**
> Our empirical findings align well with the theoretical results derived from the simplified model:
> - **Layer Freezing and Accumulated Shift**:
>   - `Proposition 4.9` suggests that freezing bottom layers reduces spurious forgetting by mitigating the accumulated shift.
>   - In `Figure 7(a)`, empirical evidence confirms that freezing more bottom layers alleviates spurious forgetting.
> - **Shift in Input Features**:
>   - `Proposition 4.7` highlights near-orthogonality between the feature shift and its principal component.
>   - This is empirically supported by the layer-wise feature visualizations in `Figure 5`. These findings demonstrate that the simplified model effectively captures the feature dynamics leading to spurious forgetting.
>
> ### **3. Precedent for Linear Models in Theoretical Analysis**
> Linear models are commonly employed for theoretical analysis due to the complexity of state-of-the-art architectures like Transformers. Examples include:
> - **Catastrophic Forgetting in Finetuning**:
>   - [R20] uses linear regression to analyze task distributional mismatch during pretraining and finetuning.
> - **Continual Learning**:
>   - Several studies [R21-25] explore continual learning theoretically using overparameterized models and linear approximations. For example:
>     - [R21] derives analytical expressions for forgetting in overparameterized models.
>     - [R22] provides generalization bounds linking forgetting severity to task dissimilarity.
>     - [R23] shows that intermediate tasks experience the highest forgetting in a student-teacher setup.
>     - [R24-25] identify regimes where dissimilar tasks are challenging, and intermediate similarity can benefit regularization.
> - **In-Context Learning (ICL)**:
>   - Linear regression is also popular for studying ICL in Transformers [R26-28]. For instance:
>     - [R26] examines Transformers’ ability to learn simple functions.
>     - [R27] investigates in-context learning through linear model approximations.
>
> Given these precedents, our use of a simplified linear model is consistent with common practices for deriving theoretical insights into complex systems like Transformers.
>
> We hope this response clarifies the rationale behind our modeling choice and its alignment with both empirical results and existing theoretical frameworks.
>
>
> [R18] A theoretical understanding of shallow vision transformers: Learning, generalization, and sample complexity (ICLR 2023)
>
> [R19] Transformers learn through gradual rank increase (NIPS 2023)
>
> [R20] Understanding Catastrophic Forgetting in Language Models via Implicit Inference (ICLR 2024)
>
> [R21] The Joint Effect of Task Similarity and Overparameterization on Catastrophic Forgetting — An Analytical Model (ICLR 2024)
>
> [R22] Generalisation guarantees for continual learning with orthogonal gradient descent (Workshop at ICML 2020)
>
> [R23] Continual learning in the teacher-student setup: Impact of task similarity (ICML 2021)
>
> [R24] Fixed design analysis of regularization-based continual learning. (CoLLA 2023)
>
> [R25] Anatomy of catastrophic forgetting: Hidden representations and task semantics (ICLR 2021)
>
> [R26] What can transformers learn in-context? a case study of simple function classes (NIPS 2022)
>
> [R27] What learning algorithm is in-context learning? investigations with linear models (ICLR 2023)
>
> [R28] Transformers as algorithms: Generalization and stability in in-context learning (ICML 2023)

---

> ### Author Response · Authors · 2024-11-23
> **Response to Reviewer RhW8 (Part 8/8)**
>
> > **Q2-2: What are the limitations of using a linear model?**
>
> **Response to Q2-2**:
> Using a stacked linear model with residual connections does not account for the non-linearities present during model training. We acknowledge this limitation and plan to extend the theoretical explanation of spurious forgetting to more complex model architectures in future work.
>
> ---
>
> > **Q3: Why isn’t exact match accuracy used as the evaluation metric for all tasks?**
>
> **Response to Q3**:
> Calculating exact match accuracy in a step-wise manner is computationally expensive. For example, in `Figure 2(a)`, Task 0 ACC and Task 1 ACC are evaluated hundreds of times. To address this, we use first-token accuracy as the metric in `Figure 2(a)`. Since first-token accuracy serves as an upper bound for exact match accuracy, its use does not alter the observed phenomena of spurious forgetting or the subsequent findings.
>
> ---
>
> > **Q4: From a model weight perspective, why is the angle computed based on the column spaces rather than directly from the weight updates?**
>
> **Response to Q4**:
> Using Singular Value Decomposition (SVD) to identify column spaces helps capture the *representative update directions that most significantly impact the output hidden states in the residual stream*. In contrast, directly computing the angle between weight updates reflects only the parameter changes, ignoring their influence on the output hidden states. This approach provides a more meaningful interpretation of the updates’ effects on the model's behavior.

---

### Author Response · Authors · 2024-11-23
**General Response to All Reviewers**

We sincerely thank the reviewers for their thoughtful feedback and constructive suggestions. Below, we summarize the key strengths acknowledged by the reviewers and outline the updates made to address their concerns.

### **Key Strengths Highlighted by Reviewers**

1. **Novel Perspective on Forgetting in Continual Learning** (`RhW8`, `zFfn`, `rjuy`)

2. **Theoretical and Experimental Support** (`zFfn`, `rjuy`)

3. **Effectiveness of the Freeze Strategy** (`RhW8`, `zFfn`, `rjuy`)

4. **Quality of Presentation** (`RhW8`, `zFfn`)

---

### **Key Updates to the Manuscript**
To address the reviewers’ concerns, we have made the following updates (highlighted in blue in the revised manuscript):

1. **Real-World Experiments**:
   Added experiments on supervised finetuning for math and code datasets to demonstrate generalizability (`Appendix I.5`).

2. **Comparisons with Advanced Methods**:
   Included comparisons to LAMOL and EWC on continual learning benchmarks (`Table 10` in `Appendix I.3.3`).

3. **Investigation of Instability in Fine-Tuning**:
   Conducted experiments analyzing spurious forgetting under different optimizers and learning rates (`Figure 12` in `Appendix G.1.4`).

4. **Expanded Related Work**:
   Added a discussion on parameter-freezing methods to situate our contributions within the broader literature (`Appendix A.3`).

5. **Additional Explanations**:
   - Clarified trajectory interpretation under data replay (`Appendix G.2`).
   - Enhanced the explanation of axes in the loss landscape figure (`Figure 3`).
   - Additional discussion of the connection between `Proposition 4.9` and the Freeze strategy (`Remark F.10`).

6. **Additional Theorectical Results**:
   - We present `Corollary F.5` and prove that freezing the bottom layers helps mitigate cumulative shift in the real-world scenario, where the orthogonal updates primarily occur in the bottom layers. ( `Assumption F.4`,`Corollary F.5`,`Remark F.6`)

We hope these revisions and additional experiments address the reviewers' concerns and further substantiate the significance of our contributions.

---

### Meta-Review · Area_Chair_pt5s · 2024-12-16

**Metareview:**

The paper investigates "spurious forgetting" in the context of continual learning for large language models (LLMs). It challenges the assumption that performance degradation in sequential tasks equates to actual knowledge loss, introducing the concept of "pseudo-forgetting," where task misalignment, rather than knowledge decay, leads to performance drops. The authors provide a theoretical framework, experiments on a synthetic biographical dataset, and analysis of model parameters to support this claim. Based on their findings, they propose a simple freezing method for the lower layers of LLMs, which mitigates spurious forgetting and improves continual learning performance.

### Strengths

- Introduces a novel and insightful distinction between spurious (pseudo) forgetting and knowledge forgetting, enhancing understanding of LLM behavior in continual learning scenarios.
- Provides a well-structured theoretical framework, supported by thorough experiments and analyses, including parameter and loss landscape studies.
- The "Freeze" approach is simple, effective, and practical, achieving competitive results while being easy to implement.
- Demonstrates clear explanations and strong presentation quality, making complex concepts accessible to a broad audience.
- Outperforms baseline methods in the specific experiments conducted, providing promising evidence for the freezing approach.

### Weaknesses

- The proposed freezing method is not novel, as parameter freezing has been widely studied in prior work (e.g., the VGG paper), and the authors fail to adequately position their contribution within this existing literature.
- Experiments rely heavily on a synthetic dataset, raising concerns about the generalizability of the findings to real-world scenarios and more diverse datasets.
- Empirical support for key theoretical assumptions (e.g., orthogonal updates causing misalignment) is limited and partly contradicted by observed results.
- Figures (e.g., Figure 3) are unclear or poorly explained, hindering interpretability and alignment with the paper's claims.
- Comparisons to baselines are weak, with only one (SEQ) real-world baseline tested, and more robust benchmarks are needed to demonstrate the effectiveness of the method.

Some concerns have been addressed by the authors during the rebuttal period.

**Additional Comments On Reviewer Discussion:**

This paper receives ratings of 8, 6, and 5. The only negative reviewer asked several insightful questions and led to several meaningful revisions. While the reviewer did not further engage after the authors’ further response. I feel the questions are mostly addressed and encourage the authors to incorporate the changes in the revision. It would also help if the authors could better position their contributions in the context of continual learning for LLMs, e.g., directions/areas discussed in [1].

[1] Shi, H., Xu, Z., Wang, H., Qin, W., Wang, W., Wang, Y., Wang, Z., Ebrahimi, S. and Wang, H., 2024. Continual learning of large language models: A comprehensive survey. arXiv preprint arXiv:2404.16789.

---

### Decision · Program_Chairs · 2025-01-22

Accept (Poster)